# Experimental visualization of water/ice phase distribution at cold start for practical-sized polymer electrolyte fuel cells

Yuki Higuchi [1] ✉, Wataru Yoshimune [1] ✉, Satoru Kato [1], Shogo Hibi[1], Daigo Setoyama [1], Kazuhisa Isegawa[1,2], Yoshihiro Matsumoto[3], Hirotoshi Hayashida[3], Hiroshi Nozaki [1], Masashi Harada [1], Norihiro Fukaya [1], Takahisa Suzuki [1], Takenao Shinohara[2] & Yasutaka Nagai [1]

The automotive industry aims to ensure the cold-start capability of polymer electrolyte fuel cells (PEFCs) for developing fuel cell electric vehicles that can be driven in cold climates. Water and ice behavior is a key factor in maintaining this capability. Previously reported methods for visualizing water and/or ice are limited to small-sized PEFCs ($< 50\ cm^2$), while fuel cell electric vehicles are equipped with larger PEFCs. Here, we developed a system using a pulsed spallation neutron beam to visualize water distribution and identify water/ice phases in practical-sized PEFCs at a cold start. The results show direct evidence of a stepwise freezing behavior inside the PEFC. The produced water initially accumulated at the center of the PEFC and then froze, followed by PEFC shutdown as freezing progressed. This study can serve as a reference to guide the development of cold-start protocols, cell design, and materials for next-generation fuel cell electric vehicles.

Automobiles, including electric vehicles, have enabled commute and travel all over the world. Fuel cell electric vehicles can be started and driven at sub-zero temperatures[1]; for example, the TOYOTA MIRAI exhibits a cold-start capability even at a low temperature of –30 °C[2]. The vehicle does not shutdown due to ice formation and blockage inside the polymer electrolyte fuel cells (PEFCs). This capability was achieved by estimating/purging the produced water content and prioritizing heat generation over power generation[2,3]. A comprehensive understanding of the water/ice behavior with direct evidence is essential to improve the cold-start capability.

Previous studies have reported the water/ice behavior inside small-sized PEFCs ($<50\ cm^2$) during power generation at sub-zero temperatures[4–16]. Inside a PEFC, a polymer electrolyte membrane separates the cathode and anode, which have a catalyst layer (CL) and gas channels in contact with a gas diffusion layer (GDL). When hydrogen and oxygen flow through the gas channels, an electrochemical reaction occurs with the generation of water as a by-product. The produced water passes through the cathode GDL to the gas channels and is drained from the PEFC. At sub-zero temperatures without the above-mentioned capability, the

produced water accumulates and freezes in PEFC, causing its shutdown[4–16]. For example, Ishikawa et al. reported that ice formation between the CL and GDL interface inhibits air supply to the catalyst and contributes to shutdown at sub-zero temperatures[14]. Sabharwal et al. performed sub-second *operando* X-ray microscopic tomography, showing the presence of water clusters inside the GDL in the temperature range of –15 to 0 °C[11]. TOYOTA MIRAI is equipped with a stack of several hundred cells, which are warmed up by self-heating during a cold start and can break through 0 °C before it becomes difficult to generate power, thus enabling a cold start from –30 °C. Previous studies have pointed out that it is important for the produced water to remain supercooled state until the freezing point is broken[17–19]. A cold-start experiment with a practical-sized PEFCs is needed to investigate the effect of self-heating on the cold-start capability.

Large-field neutron imaging measurements suitable for studying water behavior in large-sized PEFCs have been achieved with the energy-resolved neutron imaging system (the name of the instrument is RADEN) at beamline 22 in the Materials and Life Science Experimental Facility in the Japan Proton Accelerator Research Complex (J-PARC)[20,21]. We demonstrated water/ice identification through ex situ neutron imaging studies with

[1]Toyota Central R&D Labs., Inc., Nagakute, Aichi 480-1192, Japan. [2]Japan Atomic Energy Agency, Tokai, Ibaraki 319-1195, Japan. [3]Comprehensive Research Organization for Science and Society, Tokai, Ibaraki 319-1106, Japan. ✉ e-mail: e1571@mosk.tytlabs.co.jp; yoshimune@mosk.tytlabs.co.jp

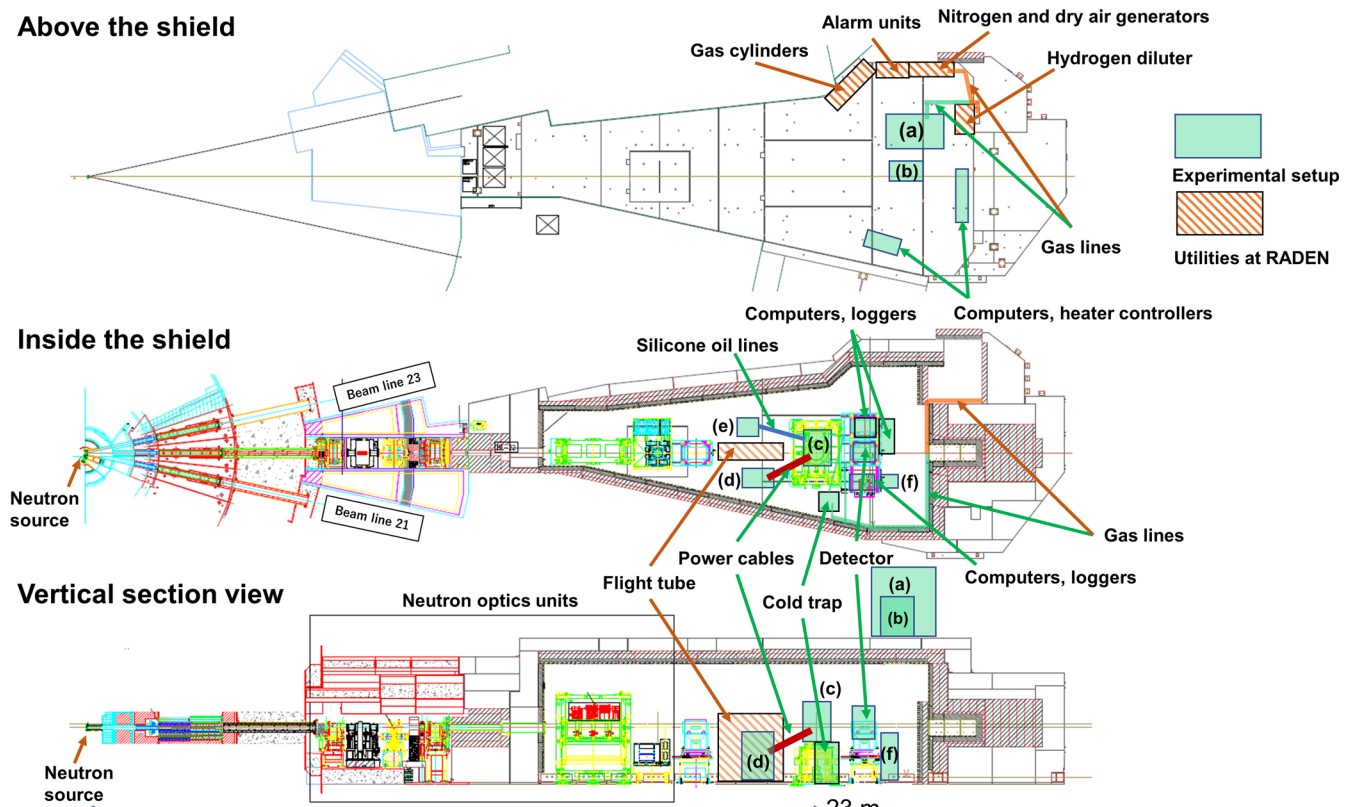

**Fig. 1 | *Operando* neutron imaging system during polymer electrolyte fuel cell power generation at the RADEN instrument.** RADEN is the name of the energy-resolved neutron imaging system. **a** Gas control unit (custom order, Enoah, Japan). **b** Chiller (RKS1503J, ORION Machinery, Japan). **c** Sample environmental control chamber (in-house). **d** Electronic load unit (KFM2150 SYSTEM, PLZ1004WS, and PLZ2004WB×2, KIKUSUI, Japan). **e** Ultra-low refrigerated/heating circulator (FP89-HL, JULABO, Germany). **f** Control and power supply unit (in-house).

a mock-up water-filled sample as a model of large-sized PEFCs[22,23]. The present study reports the development of a system for *operando* neutron imaging during power generation using large-sized PEFCs in a simulated cold climate. We conducted a cold-start experiment using our developed system to evaluate the water behavior inside a large-sized PEFC at the moment of shutdown. Water/ice identification during the cold-start experiment could not be applied due to the lack of temporal resolution. Therefore, we conducted a forced-cooling experiment to identify the water and ice phases inside the large-sized PEFC. These experiments provide deep insights into the shutdown mechanism at the cold start associated with large-sized PEFCs.

## Results

### *Operando* neutron imaging system of large-sized PEFCs at sub-zero temperatures

Figure 1 shows an *operando* neutron imaging system allowing PEFC operation from −30 to 60 °C at the RADEN instrument. The gas supply system has been installed at the exterior of beamline and connected to the large-sized PEFC on the sample stage. Temperature and electrochemical control devices were placed inside the shield of beamline. The present study used a large-sized PEFC, which is the same as in the second-generation MIRAI[17,18,24]. A PEFC consists of a polymer electrolyte membrane sandwiched between the CLs, GDLs, and flow field plates (Fig. 2a). The flow field plates supply hydrogen and air gasses in a counter-flow manner and drain the produced water from the PEFC (Fig. 2b). The cell temperatures were controlled via cooling pads to simulate a cold climate and monitored simultaneously when acquiring neutron imaging using four thermocouples (TC1–TC4) on the surface of the cathode flow field plate (Fig. 2b, c). The cooling pads were equipped with a stainless-steel cross-shaped jig to ensure fastening pressure to the PEFC (Fig. 2c, d). Pressure-sensitive paper was

used to ensure that the surface pressure was constant within the PEFC. The coolant (Fluorinert™) was flowed inside the cooling pads in the same gas flow direction (Fig. 2c, d). We have confirmed that the cooling pads are capable of cooling to a maximum of −22 °C. The PEFC and its accessories were placed in an environmental chamber with dry air flow to prevent water condensation on the surface of the cooling pads (Fig. 2e). The overall system allows PEFC power generation under a wide temperature range of −20 to 60 °C.

### Cold start at sub-zero temperatures

In this section, the water behavior inside a large-sized PEFC during a cold start from −4 °C is presented. Referring to previous papers[1–3], the power generation condition was kept as a constant-voltage mode (0.20 V) with flowing 1.4 L min⁻¹ of dry hydrogen in the anode and 2.6 L min⁻¹ of dry air in the cathode. Figure 3a shows the time-lapse of the current and cell temperature ($T_{TC1}$–$T_{TC4}$). The current increased immediately after the cold start with rising cell temperatures due to self-heating in the order of $T_{TC1} > T_{TC2} > T_{TC3} \sim T_{TC4}$, indicating that the self-heating is greater upstream of the cathode. The current once maintained 135 A and then decreased. The cell temperature dropped following the current. In the cathode upstream, the peak originating from the heat of water solidification was detected ($T_{TC1}$). The current dropped to zero (shutdown) 11 min after the cold start. Hydrogen and air continued to flow even after the shutdown, indicating that the ice formation did not block the gas supply.

Figure 3b–f show the water/ice distribution at characteristic moments inside the PEFC (see Supplementary Movie S1). In all neutron images, wavy water/ice accumulations corresponding to the anode gas channel structure[17] were observed in the lower region. The accumulation in the anode gas channel was found before the cold-start experiment. Additionally, linear

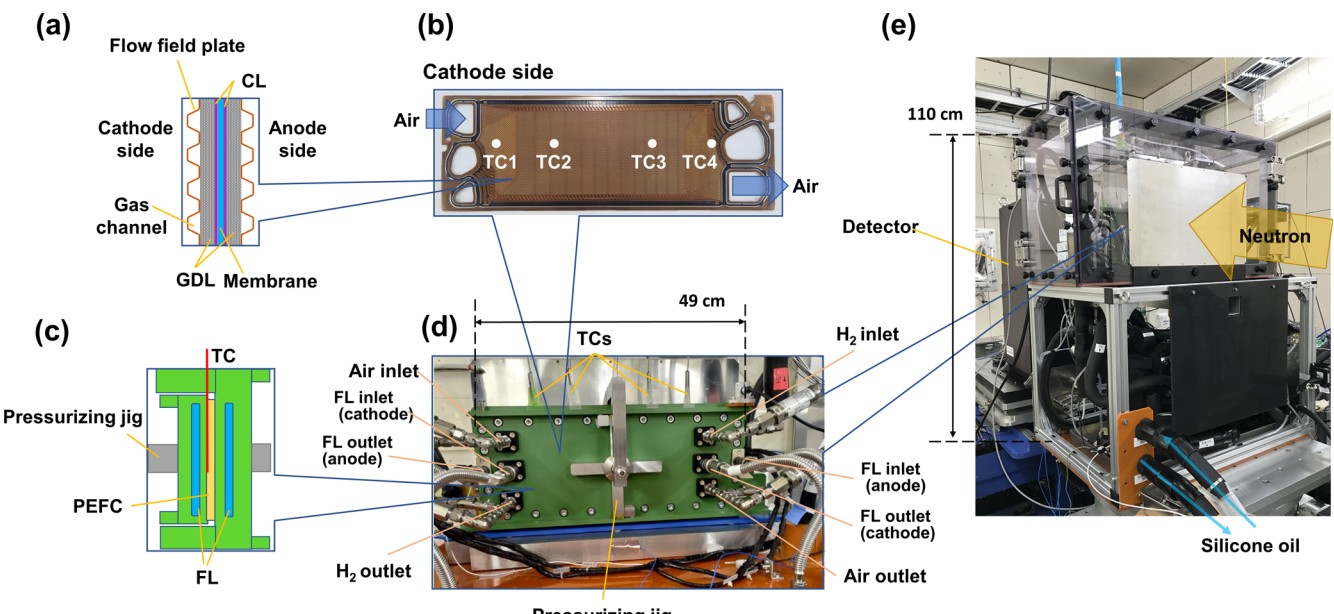

**Fig. 2 | Polymer electrolyte fuel cell power generation system. a** Cross-sectional view of polymer electrolyte fuel cell. The catalyst layer and gas diffusion layer are abbreviated as CL and GDL, respectively. **b** Photograph of flow field plates (second-generation MIRAI) with four thermocouple points (TC1–TC4). **c** Cross-sectional view of polymer electrolyte fuel cell and cooling pads. A complex pillar structure in FL flow channels is omitted for visualization. FL represents Fluorinert™. **d** Photograph of cooling pads. **e** Photograph of the environment control chamber.

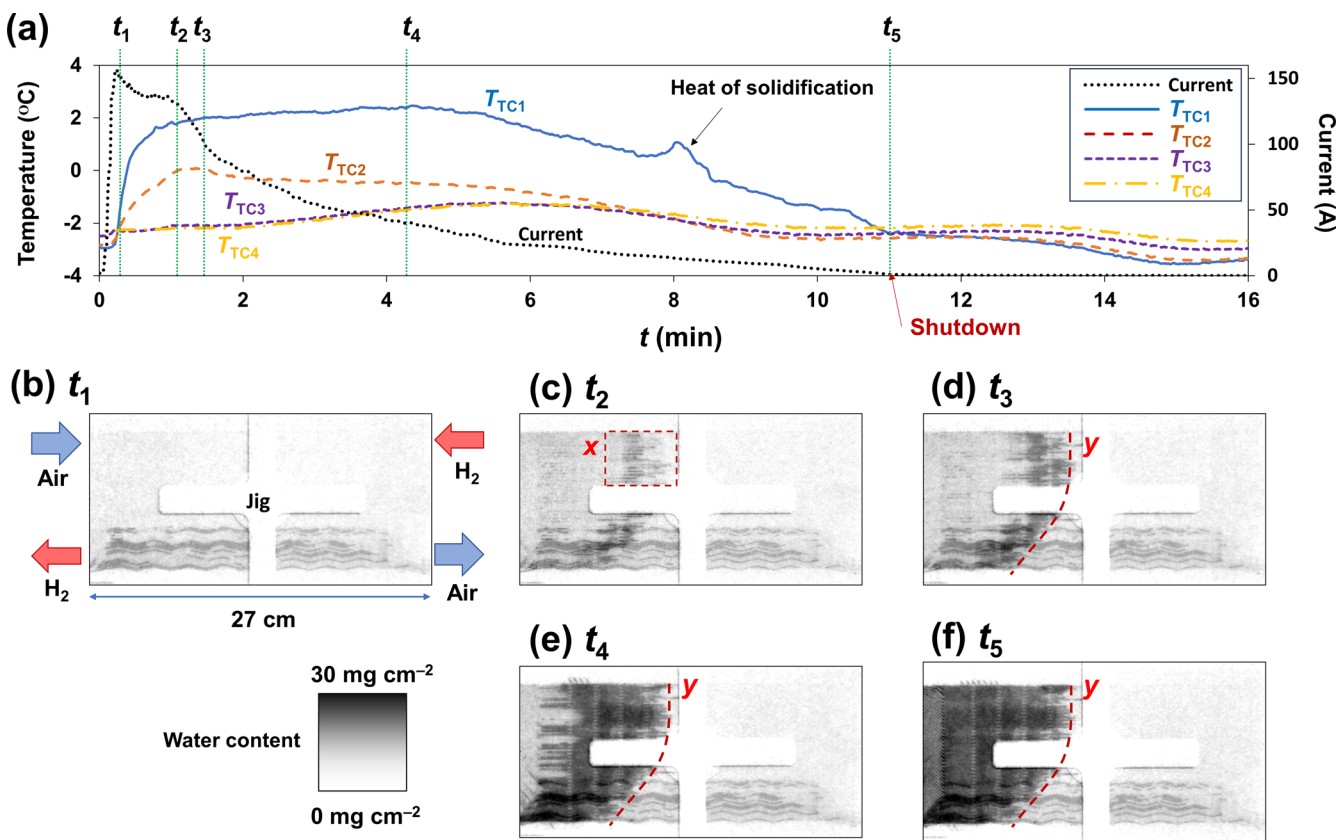

**Fig. 3 | Results of cold-start experiment. a** Time-lapse of current and cell temperature. **b–f** Selected neutron images during power generation. Water contents are shown in grayscale in every 0.3 mm square. An amount of 30 mg of water per unit area equals 300 μm of water thickness.

water/ice accumulations corresponding to the cathode gas channel[17] were observed in the *x*-region (Fig. 3c), which proceeded from the *y*-line toward the upstream (Fig. 3d, e). Subsequently, water/ice blockage occurred throughout the entire area upstream of the *y*-line (Fig. 3f).

**Forced cooling down to sub-zero temperatures**

The water/ice identification mode was applied to our *operando* neutron imaging during power generation under a forced cooling condition (down to −5 °C) to provide further insights into the shutdown mechanism at the cold

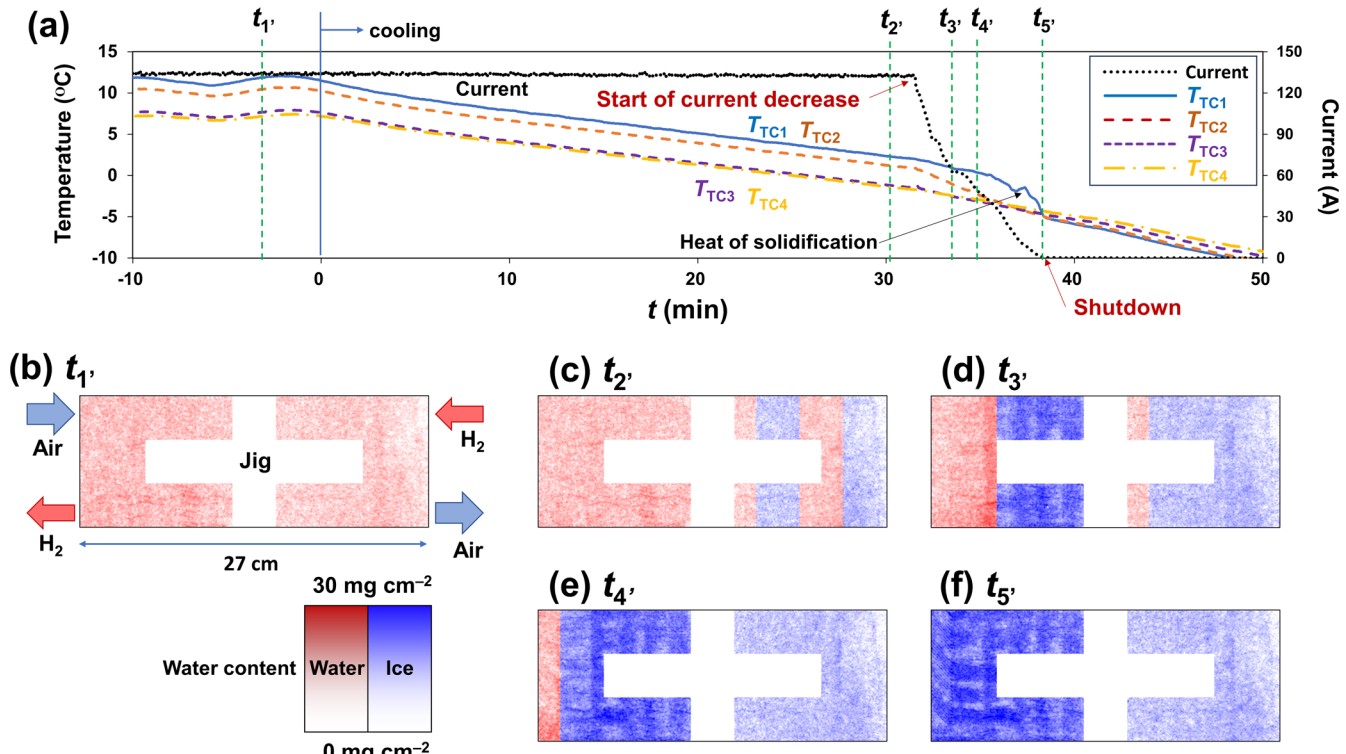

**Fig. 4 | Results of forced-cooling experiment. a** Time-lapse of current and cell temperature. **b–f** Selected neutron images during power generation. Red and blue show liquid water and ice phases, respectively. The entire area of the polymer electrolyte fuel cell, excluding the jig was divided horizontally (in every 1.7 cm width and 10.1 cm height). The color depth indicates the water/ice content in every 0.3 mm square.

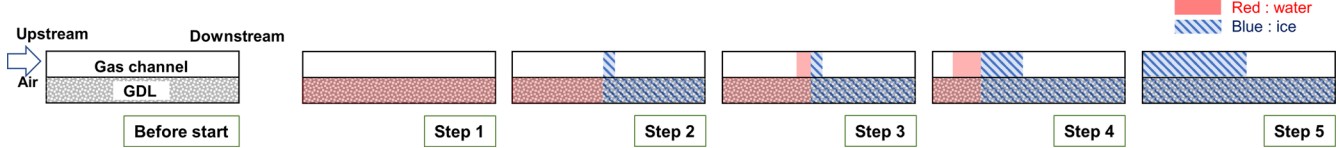

**Fig. 5 | Proposed shutdown mechanism at a cold start.** Gas diffusion layer is abbreviated as GDL.

start. The power generation conditions were constant-voltage mode (0.25 V) at 1.5 L min⁻¹ of dry hydrogen in the anode and 2.5 L min⁻¹ of dry air in the cathode. The generated PEFC was cooled at a rate of 1 °C per 3 min. The cooling rate was determined from the temporal resolution of the water/ice identification. Figure 4a shows the time-lapse of the current and temperatures ($T_{TC1}$–$T_{TC4}$). The forced-cooling experiment was started after reaching a steady state during power generation at ~10 °C. After the start of forced cooling, the cell temperature decreased linearly in the order of $T_{TC1} > T_{TC2} > T_{TC3} > T_{TC4}$, indicating that cell temperature dropped downstream of the cathode. The heat of solidification was detected in the order from downstream to upstream of the cathode. Almost simultaneously with the detection at $T_{TC1}$, the current dropped to zero. Note that hydrogen and air continued to flow even after complete freezing. Figure 4b–f show characteristic moments of the water/ice phase distribution inside the PEFC (Supplementary Movie S2). Figure 4b shows that the uniform liquid water distribution into the PEFC without any clear shape accumulated water, indicating that the water accumulated inside the GDL, not gas channels. As forced cooling proceeded, the ice phase was observed downstream of the cathode and progressed toward the upstream (Figs. 4c–f). Moreover, parallel linear ice accumulations were observed (Fig. 4d–f), indicating that ice formed in the cathode gas channels. Subsequently, the freezing area progressed toward the upstream, followed by complete freezing of the PEFC (Fig. 4f).

## Discussion

During a cold start, ice formation inhibits the gas supply to the catalyst and reduces the cold-start capability[4–16]. Our *operando* neutron imaging during the cold start revealed the stepwise water/ice behavior inside the large-sized PEFC. Additionally, the forced cooling experiment provided insights into the freezing process and the consequent shutdown mechanism (Fig. 5) due to ice formation at the cathode, with direct evidence of water/ice distributions at characteristic moments. First, produced water accumulates inside the GDL (Step 1, see Fig. 3b). Next, the water freezes from downstream, and produced water upstream moves to the gas channels and freezes (Step 2, see Figs. 3c and 4c). Subsequently, produced water accumulates upstream of the frozen zone (Step 3, see Fig. 3d–f), followed by freezing toward the upstream (Step 4, see Fig. 4e). Finally, the water freezes completely, i.e., shutdown occurs (Step 5, see Fig. 4f). This stepwise shutdown mechanism occurs because of the in-plane temperature difference peculiar to large-sized PEFCs. The current did not go down to zero at Steps 3–5. Hereafter, we call the time interval from the start of freezing to the shutdown the extended time. The PEFC upstream continues to perform power generation and self-heating during the extended time. An alternative option to prevent shutdown during a cold start is to prolong the extended time for re-thawing the ice blockage by self-heating.

In conclusion, we developed an *operando* neutron imaging system to investigate the water/ice behavior inside a large-sized PEFC at a cold start.

When we operated the PEFC for power generation at –4 °C, water/ice accumulated in the cathode GDL around the center of the PEFC. Then, the accumulated water/ice progressed toward the upstream, leading to the shutdown. The temporal resolution was insufficient for water/ice identification in the cold-start experiment. Therefore, the forced cooling experiment was carried out instead, which does not require a high temporal resolution. We conducted water/ice phase identification under a forced cooling condition where we operated the PEFC during power generation and decreased the cell temperature from ~10 °C down to –5 °C. Results show the stepwise freezing behavior from the downstream to upstream of the cathode in the large-sized PEFC. Identifying ice formation and its location in PEFCs provides an idea for protocol development of fuel-cell electric vehicles. We defined the time interval between the beginning of freezing and shutdown as the extended time. Our proposed PEFC shutdown mechanism can serve as a reference for engineers to create cold-start protocols by controlling operating conditions such as coolant/gas flow rate. The established water/ice identification technique using pulsed neutron sources is expected to be applied not only to PEFCs but also to water-containing devices such as redox flow batteries and water electrolyzers.

## Methods

### Supplementary of PEFC power generation system

The developed system for *operando* neutron imaging has been installed at the RADEN Instrument of Materials and Life Science Experimental Facility at J-PARC. The gas control unit can supply dry and wet gases: hydrogen (nitrogen) in the anode up to 10 L min⁻¹ and air (nitrogen) in the cathode up to 25(10) L min⁻¹, without condensation in the dew point range of 15–45 °C. In the temperature control system, Fluorinert™ (FC-3283, 3 M, USA) and silicone oil (KF-96-10cs, Shin-Etsu Chemical, Japan) were used as coolants. Fluorinert™ was selected for its high neutron transmittance[25], and it was circulated inside the cooling pads attached to the PEFC sides. The silicone oil was used to cool the Fluorinert™ in a heat exchanger. Together with the adoption of a multi-stage temperature control system, the environmental control chamber was equipped with Al windows and a purge line for the dry air flow to keep the dew point below –30 °C and prevent condensation on the surface of cooling pads. These contribute to the acquisition of high-quality neutron images by maintaining high neutron transmission and excluding hydrogenous elements from the neutron beam path as much as possible.

### Neutron imaging

Neutron imaging was conducted using the RADEN instrument of the Materials and Life Science Experimental Facility at J-PARC[19]. The proton beam power was 770 kW. The environmental chamber with the PEFC was placed on the sample stage of the RADEN at a distance of 23 m from the moderator, and a detector system was placed just behind it. The neutron beam was collimated using an aperture with a diameter of 26.4 mm installed in the shutter section of the beamline. The detector system consisted of a CMOS camera (ORCA Flash4.0v3, Hamamatsu Photonics K.K., Japan), an optical image intensifier (C14245-12112-A1, Hamamatsu Photonics K.K.), an optical lens (Nikkor 35 mm, Nikon, Japan) attached to the intensifier, and a 0.3 mm thick ZnO/⁶LiF scintillator screen (NDFast, Scintacor, UK). The field of view was 270 × 270 mm², and the spatial resolution was 0.3 mm. The $L/D$ value, which gives the beam divergence, was 720, where $D$ and $L$ are the aperture diameter and distance from the aperture to the detector, respectively. The neutron wavelength range was set as 0.23–0.88 nm by tuning the delay of the disk choppers. For the cold-start experiment, a sequential acquisition was performed with repeated 1 s exposures during PEFC power generation. Furthermore, for the forced-cooling experiment, neutron-energy-selected images were recorded using a wavelength-selection technique to obtain information for identifying the water/ice phases within the PEFC. Details on the wavelength-selection technique of a pulsed neutron beam have been described in our previous paper[21]. In this experiment, two neutron wavelength ranges were employed: 0.28–0.29 nm

as the short wavelength range (SW) and 0.63–0.82 nm as the long wavelength range (LW). These wavelength range conditions were switched every 5 s with an interval of 0.85 s and repeated until the shutdown at the forced-cooling experiment.

### Image correction

Image processing was performed using ImageJ[26]. As a first step, the acquired neutron images were corrected using a dark-current image. The corrected images were then divided by an image after a sufficient dry nitrogen flow inside the PEFC. The neutron attenuation caused by the produced water during PEFC power generation was only a few percent. Therefore, we corrected brightness fluctuations in the neutron images caused by other factors for accurate water quantification and water/ice phase identification. These factors include fluctuations of the incident neutron beam intensity and degradation of the detector efficiency by the high-intensity neutron irradiation.

### Water content evaluation

The spatial distribution of the water content ($d$: mg cm⁻²) was evaluated from the corrected neutron transmittance ($T$) using the calibration value (0.0042) calculated from the neutron transmittance of a water-filled quartz cell as follows:

$$d = \frac{-\ln(T)}{0.0042}. \tag{1}$$

After calibration, the image was binned both spatially and temporary into a spatial resolution of 0.26 mm (integrated width of two pixels and height of two pixels) and an interval of 20.8 s (20 integrated images) to improve the accuracy of the water content evaluation at each position.

### Water/ice phase identification

Neutron imaging data acquired at two different neutron wavelength ranges (SW and LW) were used for the processing[20,21,27–29]. An indicator for water phase identification defined as $\alpha$ is expressed as

$$\alpha = \frac{\ln(T_{LW})}{\ln(T_{SW})}. \tag{2}$$

The entire area of PEFC excluding the jig and surroundings of the PEFC was divided into 16 horizontal sections having 1.7 cm width and 10.1 cm height and accumulated temporally into an interval of ~2 min. While each section was set fairly large compared with the detector pixel for achieving sufficient statistical accuracy because of the little water content for the phase identification, its rectangular shape helped in investigating temporal change of the water/ice phase distribution along the gas flow direction. Phase identification of water and ice was conducted in the following manner. First, the identification indicator $\alpha$ for a water-filled quartz cell was calculated in both frozen ($\alpha = 1.38$) and liquid ($\alpha = 1.48$) conditions. Then, the criterion number for judging these phases was decided to be 1.43 from the mean of these values. Figure 6 shows that the identification indicator $\alpha$ crosses the threshold during the forced cooling experiment, indicating a phase transition. We decided the segmentation area (16 horizontal sections) to increase the statics for ensuring the phase identification. Consequently, a section was identified as frozen and depicted by blue color when its value for $\alpha$ was the criterion number or below, and it was identified as liquid and depicted by red color when $\alpha$ was above the criterion number. Additionally, the water/ice content $d'$ was evaluated simultaneously using the neutron transmittance at SW range. Note that the calibration value used here (0.0038) was different from that of Eq. (1) because the neutron wavelength range was not the same:

$$d' = \frac{-\ln(T_{SW})}{0.0038}. \tag{3}$$

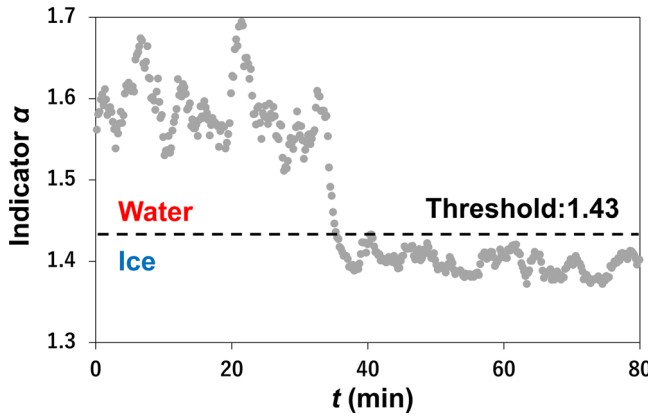

**Fig. 6 | An example of how the water/ice identification indicator changes during a forced cooling experiment.** Each point represents the average value for 2 min.

## Data availability

The data that support the findings of this study are available from the corresponding author upon reasonable request.

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

## Acknowledgements
The authors thank Tadao Ozawa, Naoki Katayama, Miwa Kanazawa, Mikiya Mori, Yumie Furuhashi, and Shota Yokoi (Toyota Central R&D Labs., Inc.) for the design and fabrication of the environmental chamber system and its accessories. This study was conducted under proposal nos. 2019L0403 and 2021B0317 as approved by J-PARC.

## Author contributions
Y.H., D.S., T. Shinohara., and Y.N. designed the research. Y.H., W.Y., S.K., S.H., D.S., K.I., Y.M., H.H., H.N., M.H., N.F., T. Suzuki., T. Shinohara. conducted experiments. Y.H. performed data analysis. Y.H. wrote the original draft. W.Y. and S.K. revised the original draft. Y.N. supervised the entire work. All authors approved the final version of the manuscript.

## Competing interests

The PEFC of TOYOTA MIRAI used for the measurements was provided by Toyota Motor Corporation. The commercial materials are identified in this document. Such identification does not imply recommendation or endorsement by the Toyota Central R&D Labs., Inc. and Toyota Motor Corporation. The authors declare no competing financial interest.
