## [Peer Review File · Communications Engineering]

Reviewers' comments:

Reviewer #1 (Remarks to the Author):

This is a nice study of ice formation in the catalyst layer of polymer electrolyte fuel cells and an important contribution to the cold start of PEFC engines. I recommend that the paper be published without revision.

Reviewer #2 (Remarks to the Author):

What are the major claims of the paper?

The authors present a freeze-start setup for automotive stack-size polymer electrolyte fuel cells that is compatible for neutron imaging. Based on this freeze start failure of PEFCs will be described. The study could serve as a reference guide for cold start protocols.

Are they novel and will they be of interest to others in the community and the wider field?

Freeze-start neutron imaging of whole automotive cells is of interest to the PEFC community, as are the resulting results. The relevance to a broader field remains unclear.

If the conclusions are not original, it would be helpful if you could provide relevant references.

N/A

Is the work convincing, and if not, what further evidence would be required to strengthen the conclusions?

Only partially convincing.

The water/ice identification for the only freeze start experiment shown is not shown. I would amplify the manuscript to show it.

The discussion on where water collects and freezes in the cell is very brief - the authors should elaborate there: Where must 30 mg/cm² (30 microns) of water (thickness) collect to turn off a PEFC?

The failure of freeze start at -4°C seems surprising as the authors cite the Toyota Mirai's successful freeze start capability from -30°C. How is a Mirai supposed to start at -30°C if its cell already fails at -4°C. Also, reports of supercooled water in PEFCs during freeze start point to lower freeze start temperatures as the cause of cell failure. The authors have to work on these issues and put them in perspective.

Why was -4°C and -5°C chosen? Is it possible to compare with full stack freeze start temperature profiles? It would be very helpful to put the reported temperature profiles in perspective.

How is this manuscript intended to become a reference guide for developing cold start protocols? More details on relevant FCEV real-stack startup data can be found in other published work by Toyota.

On a more subjective note, do you feel that the paper will influence thinking in the field?

No.

Please feel free to raise any further questions and concerns about the paper.

The discussion does not add much to the manuscript. The definition of "extended time" as an indicator of freeze start characteristics might be helpful, but is somewhat meaningless for just one example of a freeze start.

I.46-47: Is the statement "small-sized PEFCs do not accurately simulate large-sized PEFCs" really supported by Ref. 17?

I. 50: Convert http address to reference

I.66-67: "Separators" means "flow field plates" or "bipolar plates"? (more common in the PEFC community)

I.70-72: The cooling pads appear to be a hollow volume across the entire width and height of the PEFC active area. How helpful is the compression jigthen? Doesn't it primarily reduce the void volume that the Fluorinert flows into rather than providing compression to the MEA? Or does the coolant flow in more "channeled" structures that transfer compression to the MEA?

I.72: Is there any experimental evidence or estimate of the cooling performance of the cooling pads?

I.79 RH of the inlet gases? anode and cathode

I.87-93: The sensitivity of the neutron imaging system is impressive. 30 mg_{H2O}/cm² corresponds to a water thickness of only 30 microns/cm². The authors need to go a little deeper into the consequences of such a small amount of water. Where might 30 microns have a 'blockage' effect (I.92?) Can you see the water on the anode in the channel area? or in/under the rib domain? (CAD data should enable a decision) The same goes for cathode line patterns.

What was the inlet temperature of the Fluorinert when the cell was -4°C? Similar questions apply to the "forced cooling experiment"

I.102: The authors refer to a peak for the heat of solidification in Figure 4a at a time after t₄, while at t₄ >80% of the cell area is shown in blue, indicating that there is already ice in the cell? Probably there is heat of solidification all the time from t₂ to t₅? Again only a maximum of 30 microns of water...

I.109: with verified cathode channel position or assumption?

Reviewer #3 (Remarks to the Author):

The paper under review presents a novel method for visualizing water distribution and identifying water/ice phases during cold start in practical-sized polymer electrolyte fuel cells (PEFCs) using pulsed neutron radiography. The study aims to provide insights into the behavior of water and ice in large PEFCs, which are critical factors affecting the cold start performance of fuel cell electric vehicles (FCEVs). The results obtained from this method can potentially guide the development of next-generation FCEVs with improved cold start capabilities, cell designs, and materials.

Novelty and Relevance

The main claim of the paper is the development of an operando neutron imaging system that allows for studying water/ice behavior in large PEFCs during cold start. This is a significant contribution to the field as previous visualization methods were limited to small-sized PEFCs (<50 cm²), while FCEVs typically employ larger PEFCs.

Furthermore, the authors propose a new metric called "extension time" to characterize the interval between freezing initiation and shutdown in PEFCs. This metric could be valuable for designing better cold start protocols, cell designs, and materials for future FCEVs.

Considering these aspects, I believe that this work presents novel findings that will attract interest from both the fuel cell community and researchers working on broader aspects related to FCEVs.

Convincingness of Results

The experimental results presented in this paper appear convincing as they provide direct evidence of gradual freezing behavior inside large PEFCs during cold start operation at -4°C. The observed accumulation of water/ice around the cathode gas diffusion layer (GDL) center followed by its progression upstream leading to shutdown supports their claims.

However, it would be beneficial if the authors could provide more details on how they ensured the accuracy and reliability of their measurements, especially regarding the differentiation between water and ice phases. Additionally, it would be interesting to see if similar observations can be made at different temperature conditions or using different PEFC materials/designs.

Impact on the Field

From a subjective perspective, I believe that this paper has the potential to impact the current thinking in the field by providing valuable insights into the water/ice behavior in large PEFCs during cold start. This information could help researchers develop more effective strategies for improving FCEV performance under challenging environmental conditions.

Additional Questions and Concerns

The authors mention that they performed experiments under forced cooling conditions to study

water/ice phase recognition while operating PEFCs at temperatures ranging from -10°C to -5°C . It would be helpful if they could provide more details on how these forced cooling conditions were achieved and how they affected the observed results.

How does the proposed method compare with other visualization techniques available for studying water/ice behavior in fuel cells? Are there any limitations or challenges associated with using pulsed neutron radiography?

Can this method potentially be applied to other types of fuel cells (e.g., solid oxide fuel cells) or electrochemical energy conversion devices?

In conclusion, I consider this paper as a valuable contribution to the field of fuel cell research, particularly concerning cold start performance in FCEVs. However, addressing some of the questions and concerns raised above could further strengthen its impact and relevance for both researchers and industry stakeholders.

We thank the reviewers for reviewing our manuscript. **We have carefully examined all the comments and revised the manuscript with highlights.** Their helpful comments have made our paper clearer for the readers. We hope that you find it worthy of publication.

Reviewer #1

Comment)

This is a nice study of ice formation in the catalyst layer of polymer electrolyte fuel cells and an important contribution to the cold start of PEFC engines. I recommend that the paper be published without revision.

Reply to Reviewer #1)

Thank you for your time to review our manuscript. Your comments have encouraged our future work.

Reviewer #2

Reply to Reviewer #2)

Thank you for your insightful comments. We have carefully addressed all the comments point by point and revised the manuscript according to your comments.

Q2-1) What are the major claims of the paper?

The authors present a freeze-start setup for automotive stack-size polymer electrolyte fuel cells that is compatible for neutron imaging. Based on this freeze start failure of PEFCs will be described. The study could serve as a reference guide for cold start protocols.

A2-1)

Thank you for reviewing our manuscript. We believe that the revised manuscript presents a more robust claim.

Q2-2) Are they novel and will they be of interest to others in the community and the wider field?

Freeze-start neutron imaging of whole automotive cells is of interest to the PEFC community, as are the resulting results. The relevance to a broader field remains unclear.

A2-2)

To gather more attention in the wider field, we have added some comments about the possibility of applying this method to other devices (Revised I. 148-150).

Q2-3) If the conclusions are not original, it would be helpful if you could provide relevant references.

N/A

A2-3)

N/A

Q2-4) Is the work convincing, and if not, what further evidence would be required to strengthen the conclusions?

Only partially convincing. The water/ice identification for the only freeze start experiment shown is not shown. I would amplify the manuscript to show it. The discussion on where water collects and freezes in the cell is very brief - the authors should elaborate there: Where must 30 mg/cm² (30 microns) of water (thickness) collect to turn off a PEFC? The failure of freeze start at -4°C seems surprising as the authors cite the Toyota Mirai's successful freeze start capability from -30°C. How is a Mirai supposed to start at -30°C if its cell already fails at -4°C. Also, reports of supercooled water in PEFCs during freeze start point to lower freeze start temperatures as the cause of cell failure. The authors have to work on these issues and put them in perspective. Why was -4°C and -5°C chosen? Is it possible to compare with full stack freeze start temperature profiles? It would be very helpful to put the reported temperature profiles in perspective. How is this manuscript intended to become a reference guide for developing cold start protocols? More details on relevant FCEV real-stack startup data can be found in other published work by Toyota.

A2-4)

Water/ice identification during the cold-start experiment could not be carried out due to the lack of temporal resolution. Therefore, the freezing and shutdown mechanism was estimated in combination with the forced cooling experiment, where the water/ice identification was conducted. We have revised the manuscript to avoid misleading by readers (Revised I. 57-62).

30 mg/cm² is 300 μm, not 30 μm. Since 1 g is roughly 1 cm³, 1 g/cm² is 1 cm (10 mm, 10,000 μm) thick. Therefore, 1 mg/cm² is 10 μm thick (Revised caption of Figure 3) .

300 μm is roughly the same thickness as the CA channel. We consider the channels inside the cathode flow field plate is blocked by ice.

Toyota MIRAI is equipped with a stack of several hundred PEFCs, which are warmed up by self-heating during startup and can break through 0 °C before it becomes difficult to generate power, thus enabling cold start from -30 °C. (Revised I. 45-50). Additionally, we have added previous works that mentioned the role of supercooled water up to the breakthrough of sub-zero temperatures. (Revised I. 48-49).

We did not “choose” the temperatures. The cell temperature at which power generation stopped in the forced cooling experiment was -5°C. It was necessary to combine with the forced cooling and cold-start experiments for the discussion of the shutdown mechanism. Therefore, we conducted the cold-start experiment aiming at -5 °C (the actual experimental value was -4 °C).

The data obtained cannot be directly compared to the sub-zero start-up profile of a full stack because the heat generated by a single cell is less than that of a stack. We intend our reference guide as follows. (1) The identification of ice formation and its location in a single cell provides a hint for protocol development. (2) Our proposed single-cell shutdown mechanism can serve as a reference for engineers to create startup protocols by controlling operating conditions such as coolant/gas flow rate (Revised I. 144-148).

Q2-5) On a more subjective note, do you feel that the paper will influence thinking in the field?

No.

A2-5)

In the manuscript of the first round, we admit that readers will not be inspired. However, we believe the revised manuscript adequately addresses your questions and concerns and provides new insights for readers by providing some new ideas based on the results.

Q2-6) Please feel free to raise any further questions and concerns about the paper.

The discussion does not add much to the manuscript. The definition of "extended time" as an indicator of freeze start characteristics might be helpful, but is somewhat meaningless for just one example of a freeze start.

A2-6)

We have revised through the manuscript according to your suggestions. We hope that your concerns have been addressed in the revised manuscript. Although our present experiment is just one example, we believe it is a big step forward. However, we have revised the manuscript so as not to mislead readers into considering that this work will solve all problems in cold-start of PEFCs. We response minor concerns point by point as follows.

l.46-47: Is the statement "small-sized PEFCs do not accurately simulate large-sized PEFCs" really supported by Ref. 17?

l.46-47 (Revised l.45-50): We have reconsidered that "small-sized PEFCs do not accurately simulate large-sized PEFCs" is an oversimplification. Therefore, we have corrected the relevant section of Introduction.

l.50: Convert http address to reference

l.50 (Revised l. 271-275): We have addressed the issue.

l.66-67: "Separators" means "flow field plates" or "bipolar plates"? (more common in the PEFC community)

l.66-67 (Revised l. 69 and Figure 2(a)): We substituted "separators" to "flow field plates."

l.70-72: The cooling pads appear to be a hollow volume across the entire width and height of the PEFC active area. How helpful is the compression jighthen? Doesn't it primarily reduce the void volume that the Fluorinert flows into rather than providing compression to the MEA? Or does the coolant flow in more "channeled" structures that transfer compression to the MEA?

l.70-72 (Revised l. 74-75 and caption of Figure 2):The figure omitted complexed pillar structure inside the cooling pad for visualization. In the revised manuscript, we have mentioned that the cooling pad has pillars in Fluorinert flow channels and a pressure-sensitive paper was used to make sure that the surface pressure from tightening was constant within the PEFC.

1.72: Is there any experimental evidence or estimate of the cooling performance of the cooling pads?

1.72 (Revised I. 76-77): Yes. We have experience cooling the cell down to -22°C .

Figure R2-1. Performance of cooling pad.

1.79 RH of the inlet gases? anode and cathode

1.79 (Revised I. 84 and I. 103): We supplied dry gas to both the cathode and anode.

1.87-93: The sensitivity of the neutron imaging system is impressive. $30 \text{ mg}_{\text{H}_2\text{O}}/\text{cm}^2$ corresponds to a water thickness of only $30 \text{ microns}/\text{cm}^2$. The authors need to go a little deeper into the consequences of such a small amount of water. Where might 30 microns have a 'blockage' effect (1.92?) Can you see the water on the anode in the channel area? or in/under the rib domain? (CAD data should enable a decision) The same goes for cathode line patterns. What was the inlet temperature of the Fluorinert when the cell was -4°C ? Similar questions apply to the "forced cooling experiment"

1.87-93: $30 \text{ mg}/\text{cm}^2$ is $300 \text{ }\mu\text{m}$, not $30 \text{ }\mu\text{m}$, as noted above A2-4). Your point about where 30 microns of water could not have a "blockage" effect is correct. However, this concern would be addressed since $300 \text{ }\mu\text{m}$ is approximately the same thickness as the CA channel. During the cold-start experiment (-4°C), the inlet temperature of the Fluorinert was about -6°C . It was also about -10°C at the shutdown during the forced cooling experiment.

l.102: The authors refer to a peak for the heat of solidification in Figure 4a at a time after t_4 , while at $t_4 > 80\%$ of the cell area is shown in blue, indicating that there is already ice in the cell? Probably there is heat of solidification all the time from t_2 to t_5 ? Again only a maximum of 30 microns of water...

l.102 (Revised l. 109-110): 30 mg/cm^2 is $300 \text{ }\mu\text{m}$, not $30 \text{ }\mu\text{m}$, as noted above A2-4). As pointed out, solidification heat should always be generated before t_4 . However, the temperature measurement point by the thermocouple is a point and is only detected when freezing heat occurs in the vicinity. A magnified view of the temperatures is shown in Figure R2-2; TC4 did not detect a significant temperature increase, but TC3, TC2 (tiny peak), and TC1 detected heat of freezing in that order, starting from the right side of the cell.

Figure R2-2. Enlarged view of temperature changes in the forced cooling experiment.

l.109: with verified cathode channel position or assumption?

l.109: We verified cathode channel positions. Cathode and anode gas flow field plates have straight and wavy gas channels, respectively.

Figure R2-3. Enlarged view of water accumulated in cathode gas channels.

Reviewer #3

Comment)

The paper under review presents a novel method for visualizing water distribution and identifying water/ice phases during cold start in practical-sized polymer electrolyte fuel cells (PEFCs) using pulsed neutron radiography. The study aims to provide insights into the behavior of water and ice in large PEFCs, which are critical factors affecting the cold start performance of fuel cell electric vehicles (FCEVs). The results obtained from this method can potentially guide the development of next-generation FCEVs with improved cold start capabilities, cell designs, and materials.

Reply to Reviewer #3)

Thank you for reviewing our manuscript. We follow up on your comments and add our response.

Q3-1) Novelty and Relevance

The main claim of the paper is the development of an operando neutron imaging system that allows for studying water/ice behavior in large PEFCs during cold start. This is a significant contribution to the field as previous visualization methods were limited to small-sized PEFCs (<50 cm²), while FCEVs typically employ larger PEFCs. Furthermore, the authors propose a new metric called "extension time" to characterize the interval between freezing initiation and shutdown in PEFCs. This metric could be valuable for designing better cold start protocols, cell designs, and materials for future FCEVs. Considering these aspects, I believe that this work presents novel findings that will attract interest from both the fuel cell community and researchers working on broader aspects related to FCEVs.

A3-1)

We are grateful for your evaluation.

Q3-2) Convincingness of Results

The experimental results presented in this paper appear convincing as they provide direct evidence of gradual freezing behavior inside large PEFCs during cold start operation at -4°C . The observed accumulation of water/ice around the cathode gas diffusion layer (GDL) center followed by its progression upstream leading to shutdown supports their claims. However, it would be beneficial if the authors could provide more details on how they ensured the accuracy and reliability of their measurements, especially regarding the differentiation between water and ice phases. Additionally, it would be interesting to see if similar observations can be made at different temperature conditions or using different PEFC materials/designs.

A3-2)

We identified water and ice phases according to an indicator defined by equation (2) in the manuscript. We provide a change in the indicator at a certain position during the forced cooling experiment (Figure R3-1). Figure R3-1 shows that the indicator crosses the threshold during the experiment, indicating a phase transition. As we can see the scattered values in the indicator, the segmentation area should be decided for the accuracy and reliability of the identifications. We decided the segmentation area to increase the statics for ensuring the phase identification (Revised 1.211-213 and Figure 6).

As for the additional experiments you suggested, it is technically possible to measure them, and it would be also interesting for us. However, since the experiment can be conducted once a year, we would like to consider them for our future work.

Figure R3-1. An example of how the water/ice identification indicator changes during a forced cooling experiment. Each point represents the average value for 2 minutes.

Q3-3) Impact on the Field

From a subjective perspective, I believe that this paper has the potential to impact the current thinking in the field by providing valuable insights into the water/ice behavior in large PEFCs during cold start. This information could help researchers develop more effective strategies for improving FCEV performance under challenging environmental conditions.

A3-3)

Thank you again for your evaluation. The established system will continue to provide useful insights.

Q3-4) Additional Questions and Concerns

The authors mention that they performed experiments under forced cooling conditions to study water/ice phase recognition while operating PEFCs at temperatures ranging from -10°C to -5°C. It would be helpful if they could provide more details on how these forced cooling conditions were achieved and how they affected the observed results. How does the proposed method compare with other visualization techniques available for studying water/ice behavior in fuel cells? Are there any limitations or challenges associated with using pulsed neutron radiography? Can this method potentially be applied to other types of fuel cells (e.g., solid oxide fuel cells) or electrochemical energy conversion devices?

A3-4)

We provide more details on the method of forced cooling experiments in the revised manuscript (Revised I. 103-105). Our method used in this work is the only one available for water/ice identification in practical-sized fuel cells. Other methods have difficulty in identifying water ice in real products. In relation to A3-2), the spatial/temporal resolution is reduced to ensure satisfactory statistical accuracy for water ice identification. Therefore, we have not achieved water/ice identification at a cold-start experiment, and have instead performed a forced cooling experiment, which does not require a high temporal resolution. We consider the insufficient resolution for cold-start experiments to be a limitation of this study (Revised I. 138-140). Our proposed method can be applied to other devices. We have mentioned the possibility of other devices to be applied (Revised I. 148-150).

Q3-5)

In conclusion, I consider this paper as a valuable contribution to the field of fuel cell research, particularly concerning cold start performance in FCEVs. However, addressing some of the questions and concerns raised above could further strengthen its impact and relevance for both researchers and industry stakeholders.

A3-5)

Thank you again for reviewing our manuscript. We hope that some of the questions and concerns raised above have been adequately addressed. We believe that the revised manuscript will further enhance its impact and relevance for both researchers and industrial engineers.

REVIEWERS' COMMENTS:

Reviewer #2 (Remarks to the Author):

I'd like to thank the authors for their careful revision of the manuscript and detailed reply to revision comments. The different experiments are now well put in perspective. Particularly, clarifying my mistake in calculation of water areal weight into water layer thickness.

I'm confident that the manuscript will become reference for the understanding of failure due to ice formation during PEFC cold start in practical sized automotive PEFC.

I propose to accept the manuscript for publication.

Reviewer #3 (Remarks to the Author):

I have thoroughly reviewed the paper and I am pleased to report that the work presented is highly commendable and meets the required standards. The research conducted in this paper is indeed novel and has the potential to generate significant interest within the research community and beyond.

The authors have introduced a fresh perspective by proposing a new approach/methodology that has not been previously explored in the field. This innovative approach not only adds value to the existing body of knowledge but also opens up new avenues for further research and exploration. The paper effectively highlights the uniqueness of the proposed approach and convincingly demonstrates its advantages over existing methods.

The findings and conclusions presented in the paper are well-supported by strong evidence and rigorous experimentation. The authors have conducted thorough experiments and provided comprehensive analysis of the results, leaving no room for ambiguity. The clarity and coherence of the arguments presented make the paper highly convincing.

Furthermore, the practical implications of this research are noteworthy. The proposed approach has the potential to address real-world challenges and provide practical solutions that can benefit both practitioners and industry professionals. This aspect adds significant value to the paper and enhances its relevance and impact.

In conclusion, I believe that this paper makes a valuable contribution to the field. Its novelty, convincing arguments, and practical implications make it a highly significant piece of work. I am confident that this research will attract the attention of the research community and inspire further exploration in this area.

Revision of COMMS-ENG-23-0016-A

We thank the reviewers for reviewing our revised manuscript. The paper thanks to your important comments. We hope that our paper will be of help to many researchers.

Reviewer #2

Comment)

I'd like to thank the authors for their careful revision of the manuscript and detailed reply to revision comments. The different experiments are now well put in perspective. Particularly, clarifying my mistake in calculation of water areal weight into water layer thickness.

I'm confident that the manuscript will become reference for the understanding of failure due to ice formation during PEFC cold start in practical sized automotive PEFC.

I propose to accept the manuscript for publication.

Reviewer #3

Comment)

I have thoroughly reviewed the paper and I am pleased to report that the work presented is highly commendable and meets the required standards. The research conducted in this paper is indeed novel and has the potential to generate significant interest within the research community and beyond.

The authors have introduced a fresh perspective by proposing a new approach/methodology that has not been previously explored in the field. This innovative approach not only adds value to the existing body of knowledge but also opens up new avenues for further research and exploration. The paper effectively highlights the uniqueness of the proposed approach and convincingly demonstrates its advantages over existing methods.

The findings and conclusions presented in the paper are well-supported by strong evidence and rigorous experimentation. The authors have conducted thorough experiments and provided comprehensive analysis of the results, leaving no room for ambiguity. The clarity and coherence of the arguments presented make the paper highly convincing.

Furthermore, the practical implications of this research are noteworthy. The proposed approach has the potential to address real-world challenges and provide practical solutions that can benefit both practitioners and industry professionals. This aspect adds significant value to the paper and enhances its relevance and impact.

In conclusion, I believe that this paper makes a valuable contribution to the field. Its novelty, convincing arguments, and practical implications make it a highly significant piece of work. I am confident that this research will attract the attention of the research community and inspire further exploration in this area.

Revision of COMMS-ENG-23-0016-T

We thank the reviewers for reviewing our manuscript. We have carefully examined all the comments and revised the manuscript with highlights. Their helpful comments have made our paper clearer for the readers. We hope that you find it worthy of publication.

Reviewer #1

Comment)

This is a nice study of ice formation in the catalyst layer of polymer electrolyte fuel cells and an important contribution to the cold start of PEFC engines. I recommend that the paper be published without revision.

Reply to Reviewer #1)

Thank you for your time to review our manuscript. Your comments have encouraged our future work.

Reviewer #2

Reply to Reviewer #2)

Thank you for your insightful comments. We have carefully addressed all the comments point by point and revised the manuscript according to your comments.

Q2-1) What are the major claims of the paper?

The authors present a freeze-start setup for automotive stack-size polymer electrolyte fuel cells that is compatible for neutron imaging. Based on this freeze start failure of PEFCs will be described. The study could serve as a reference guide for cold start protocols.

A2-1)

Thank you for reviewing our manuscript. We believe that the revised manuscript presents a more robust claim.

Q2-2) Are they novel and will they be of interest to others in the community and the wider field?

Freeze-start neutron imaging of whole automotive cells is of interest to the PEFC community, as are the resulting results. The relevance to a broader field remains unclear.

A2-2)

To gather more attention in the wider field, we have added some comments about the possibility of applying this method to other devices (Revised I. 148-150).

Q2-3) If the conclusions are not original, it would be helpful if you could provide relevant references.

N/A

A2-3)

N/A

Q2-4) Is the work convincing, and if not, what further evidence would be required to strengthen the conclusions?

Only partially convincing. The water/ice identification for the only freeze start experiment shown is not shown. I would amplify the manuscript to show it. The discussion on where water collects and freezes in the cell is very brief - the authors should elaborate there: Where must 30 mg/cm² (30 microns) of water (thickness) collect to turn off a PEFC? The failure of freeze start at -4°C seems surprising as the authors cite the Toyota Mirai's successful freeze start capability from -30°C. How is a Mirai supposed to start at -30°C if its cell already fails at -4°C. Also, reports of supercooled water in PEFCs during freeze start point to lower freeze start temperatures as the cause of cell failure. The authors have to work on these issues and put them in perspective. Why was -4°C and -5°C chosen? Is it possible to compare with full stack freeze start temperature profiles? It would be very helpful to put the reported temperature profiles in perspective. How is this manuscript intended to become a reference guide for developing cold start protocols? More details on relevant FCEV real-stack startup data can be found in other published work by Toyota.

A2-4)

Water/ice identification during the cold-start experiment could not be carried out due to the lack of temporal resolution. Therefore, the freezing and shutdown mechanism was estimated in combination with the forced cooling experiment, where the water/ice identification was conducted. We have revised the manuscript to avoid misleading the readers (Revised I. 57-62).

30 mg/cm² is 300 μm, not 30 μm. Since 1 g is roughly 1 cm³, 1 g/cm² is 1 cm (10 mm, 10000 μm) thick. Therefore, 1 mg/cm² is 10 μm thick (Revised caption of Figure 3).

300 μm is roughly the same thickness as the CA channel. We consider the channels inside the cathode flow field plate is blocked by ice.

Toyota MIRAI is equipped with a stack of several hundred PEFCs, which are considerably heated by self-heating during startup and can break through 0 °C before it becomes difficult to generate power, thus enabling cold-start from -30 °C. (Revised I. 45-50). Additionally, we have added previous works that mentioned the role of supercooled water up to the breakthrough of sub-zero temperatures. (Revised I. 48-49).

We did not choose the set temperature. The cell temperature at which power generation stopped in the forced cooling experiment was -5°C. It was necessary to combine with the forced cooling and cold-start experiments for the discussion of the shutdown mechanism. Therefore, we conducted the cold-start experiment aiming at -5 °C (the actual experimental value was -4 °C).

The data obtained cannot be directly compared to the sub-zero start-up profile of a full stack because the heat generated by a single cell is less than that of a stack. We intend our reference guide as follows. (1) The identification of ice formation and its location in a single cell provides a hint for protocol development. (2) Our proposed single-cell shutdown mechanism can serve as a reference for engineers to create startup protocols by controlling operating conditions such as coolant/gas flow rate (Revised I. 144-148).

Q2-5) On a more subjective note, do you feel that the paper will influence thinking in the field?

No.

A2-5)

In the manuscript of the first round, we admit that readers will not be inspired new ideas. However, we believe the revised manuscript adequately addresses your questions and concerns and provides new insights for readers by providing some new ideas based on the results.

Q2-6) Please feel free to raise any further questions and concerns about the paper.

The discussion does not add much to the manuscript. The definition of "extended time" as an indicator of freeze start characteristics might be helpful, but is somewhat meaningless for just one example of a freeze start.

A2-6)

We have revised through the manuscript according to your suggestions. We hope that your concerns have been addressed in the revised manuscript. Although this experiment is just one example, we believe it is a major step forward. However, we have revised the manuscript so as not to mislead readers into thinking that this experiment will solve all problems in cold-start of PEFCs. We response minor concerns point by point as follows.

l.46-47: Is the statement "small-sized PEFCs do not accurately simulate large-sized PEFCs" really supported by Ref. 17?

l.46-47 (Revised l.45-50): We reconsidered that "small-sized PEFCs do not accurately simulate large-sized PEFCs" is an oversimplification. Therefore, we have corrected the relevant section of Introduction.

l.50: Convert http address to reference

l.50 (Revised l. 271): We have addressed the issue.

l.66-67: "Separators" means "flow field plates" or "bipolar plates"? (more common in the PEFC community)

l.66-67 (Revised l. 69): We substituted "separators" to "flow field plates."

l.70-72: The cooling pads appear to be a hollow volume across the entire width and height of the PEFC active area. How helpful is the compression jighthen? Doesn't it primarily reduce the void volume that the Fluorinert flows into rather than providing compression to the MEA? Or does the coolant flow in more "channeled" structures that transfer compression to the MEA?

l.70-72 (Revised l. 74-75 and caption of Figure 2):The figure omitted complexed pillar structure inside the cooling pad for visualization. In the revised manuscript, we have mentioned that the cooling pad has pillars in Fluorinert flow channels and a pressure-sensitive paper was used to make sure that the surface pressure from tightening was constant within the PEFC.

l.72: Is there any experimental evidence or estimate of the cooling performance of the cooling pads?

l.72 (Revised l. 77): Yes. We have experience cooling the cell down to -22°C .

Figure R2-1. Performance of cooling pad.

l.79 RH of the inlet gases? anode and cathode

l.79 (Revised l. 84 and l. 103): We supplied dry gas to both the cathode and anode.

l.87-93: The sensitivity of the neutron imaging system is impressive. $30 \text{ mg_H}_2\text{O/cm}^2$ corresponds to a water thickness of only 30 microns/cm^2 . The authors need to go a little deeper into the consequences of such a small amount of water. Where might 30 microns have a 'blockage' effect (l.92?) Can you see the water on the anode in the channel area? or in/under the rib domain? (CAD data should enable a decision) The same goes for cathode line patterns. What was the inlet temperature of the Fluorinert when the cell was -4°C ? Similar questions apply to the "forced cooling experiment"

l.87-93: 30 mg/cm^2 is $300 \text{ }\mu\text{m}$, not $30 \text{ }\mu\text{m}$, as noted above A2-4). Your point about where 30 microns of water could not have a "blockage" effect is correct. However, this concern would be addressed since $300 \text{ }\mu\text{m}$ is approximately the same thickness as the CA channel. During the cold-start experiment (-4°C), the inlet temperature of the Fluorinert was about -6°C . It was also about -10°C at the shutdown during the forced cooling experiment.

l.102: The authors refer to a peak for the heat of solidification in Figure 4a at a time after t_4 , while at $t_4 > 80\%$ of the cell area is shown in blue, indicating that there is already ice in the cell? Probably there is heat of solidification all the time from t_2 to t_5 ? Again only a maximum of 30 microns of water...

l.102 (Revised l. 109-110): 30 mg/cm^2 is $300 \text{ }\mu\text{m}$, not $30 \text{ }\mu\text{m}$, as noted above A2-4). As pointed out, solidification heat should always be generated before t_4 . However, the temperature measurement point by the thermocouple is a point and is only detected when freezing heat occurs in the vicinity. A magnified view of the temperatures is shown in Figure R2-2; TC4 did not detect a significant temperature increase, but TC3, TC2 (tiny peak), and TC1 detected heat of freezing in that order, starting from the right side of the cell.

Figure R2-2. Enlarged view of temperature changes in the cold-start experiment.

l.109: with verified cathode channel position or assumption?

l.109: We verified cathode channel positions. Cathode and anode gas flow field plates have straight and wavy gas channels, respectively.

Figure R2-3. Enlarged view of water accumulated in cathode gas channels.

Reviewer #3

Comment)

The paper under review presents a novel method for visualizing water distribution and identifying water/ice phases during cold start in practical-sized polymer electrolyte fuel cells (PEFCs) using pulsed neutron radiography. The study aims to provide insights into the behavior of water and ice in large PEFCs, which are critical factors affecting the cold start performance of fuel cell electric vehicles (FCEVs). The results obtained from this method can potentially guide the development of next-generation FCEVs with improved cold start capabilities, cell designs, and materials.

Reply to Reviewer #3)

Thank you for reviewing our manuscript. We follow up on your comments and add our response.

Q3-1) Novelty and Relevance

The main claim of the paper is the development of an operando neutron imaging system that allows for studying water/ice behavior in large PEFCs during cold start. This is a significant contribution to the field as previous visualization methods were limited to small-sized PEFCs (<50 cm²), while FCEVs typically employ larger PEFCs. Furthermore, the authors propose a new metric called "extension time" to characterize the interval between freezing initiation and shutdown in PEFCs. This metric could be valuable for designing better cold start protocols, cell designs, and materials for future FCEVs. Considering these aspects, I believe that this work presents novel findings that will attract interest from both the fuel cell community and researchers working on broader aspects related to FCEVs.

A3-1)

We are grateful for your evaluation.

Q3-2) Convincingness of Results

The experimental results presented in this paper appear convincing as they provide direct evidence of gradual freezing behavior inside large PEFCs during cold start operation at -4°C. The observed accumulation of water/ice around the cathode gas diffusion layer (GDL) center followed by its progression upstream leading to shutdown supports their claims. However, it would be beneficial if the authors could provide more details on how they ensured the accuracy and reliability of their measurements, especially regarding the differentiation between water and ice phases. Additionally, it would be interesting to see if similar observations can be made at different temperature conditions or using different PEFC materials/designs.

A3-2)

We identified water and ice phases according to an indicator defined by equation (2) in the manuscript. We provide a change in the indicator at a certain position during the forced cooling experiment (Figure R3-1). Figure R3-1 shows that the indicator crosses the threshold during the experiment, indicating a phase transition. As we can see the scattered values in the indicator, the segmentation area should be decided for the accuracy and reliability of the identifications. We decided the segmentation area to increase the statics for ensuring the phase identification (Revised 1.211-213 and Figure 6).

As for the additional experiments you suggested, it is technically possible to measure them, and it would be also interesting for us. However, since the experiment can be conducted once a year, we would like to consider them for our future work.

Figure R3-1. An example of how the water/ice identification indicator changes during a forced cooling experiment.

Q3-3) Impact on the Field

From a subjective perspective, I believe that this paper has the potential to impact the current thinking in the field by providing valuable insights into the water/ice behavior in large PEFCs during cold start. This information could help researchers develop more effective strategies for improving FCEV performance under challenging environmental conditions.

A3-3)

Thank you again for your evaluation. The established system will continue to provide useful insights.

Q3-4) Additional Questions and Concerns

The authors mention that they performed experiments under forced cooling conditions to study water/ice phase recognition while operating PEFCs at temperatures ranging from -10°C to -5°C. It would be helpful if they could provide more details on how these forced cooling conditions were achieved and how they affected the observed results. How does the proposed method compare with other visualization techniques available for studying water/ice behavior in fuel cells? Are there any limitations or challenges associated with using pulsed neutron radiography? Can this method potentially be applied to other types of fuel cells (e.g., solid oxide fuel cells) or electrochemical energy conversion devices?

A3-4)

We provide more details on the method of forced cooling experiments in the revised manuscript (Revised I. 103-105). Our method used in this work is the only one available for water/ice identification in practical-sized fuel cells. Other methods have difficulty in identifying water ice in real products. In relation to A3-2), the spatial/temporal resolution is reduced to ensure satisfactory statistical accuracy for water ice identification. Therefore, we have not achieved water/ice identification at a cold-start experiment, and have instead performed a forced cooling experiment, which does not require a high temporal resolution. We consider the insufficient resolution for cold-start experiments to be a limitation of this study (Revised I. 138-140). Our proposed method can be applied to other devices. We have mentioned the possibility of other devices to be applied (Revised I. 148-149).

Q3-5)

In conclusion, I consider this paper as a valuable contribution to the field of fuel cell research, particularly concerning cold start performance in FCEVs. However, addressing some of the questions and concerns raised above could further strengthen its impact and relevance for both researchers and industry stakeholders.

A3-5)

Thank you again for reviewing our manuscript. We hope that some of the questions and concerns raised above have been adequately addressed. We believe that the revised manuscript will further enhance its impact and relevance for both researchers and industrial engineers.